# Tailoring the Performance of Graphene Aerogels for Oil/Organic Solvent Separation by 1-Step Solvothermal Approach

**DOI:** 10.3390/nano9081077

**Published:** 2019-07-26

**Authors:** Alina Pruna, Alfonso C. Cárcel, Arturo Barjola, Adolfo Benedito, Enrique Giménez

**Affiliations:** 1Instituto de Tecnología de Materiales, Universitat Politècnica de València (UPV), Camino de Vera s/n, 46022 Valencia, Spain; 2Center for Surface Science and Nanotechnology, University Politehnica of Bucharest, 316 Splaiul Independentei, 060042 Bucharest, Romania; 3Instituto Tecnológico del Plástico (AIMPLAS), 46980 Paterna, Valencia, Spain

**Keywords:** graphene oxide, ascorbic acid, aerogel, ice-templating: oil absorption, organic solvent absorption

## Abstract

Ultra-light eco-friendly graphene oxide (GO)-based aerogels are reported by simple one-step solvothermal self-assembly. The effect of varying parameters such as C/O ratio of GO; reducing agent amount; temperature; and duration on the properties of the aerogels was studied. The structural and vibrational features and hydrophobic surface properties of the obtained aerogels were obtained by XRD; FTIR; XPS; Raman; SEM; and contact angle measurements. The effect of synthesis conditions on the engine oil and organic solvent absorption properties was assessed. The results indicated that the lower the C/O ratio of GO, the better the absorption properties, with the best performance for oil uptake reaching 86 g g^−1^. The obtained results indicate the approach based on ice-templating and the tailoring of oxygen content in GO make the resulting aerogels potential candidates for use in oil spill and organic solvent treatments.

## 1. Introduction

Serious environmental concerns have been generated by oil spills in the sea in the last few years. Moreover, accidental spills of organic solvents add to the magnitude of the issue. A wide variety of methods for environmental remediation from such pollutants including absorption, photodegradation and in-situ burning have been developed [1,2,3]. The most interesting method considers the use of physical absorption thanks to its high efficiency, simplicity and green feature. Since the conventional absorbents show drawbacks such as poor separation ability or secondary pollution, the development of low-cost eco-friendly absorbents with high selectivity, recyclability and absorption capacity has attracted increased focus.

Three-dimensional (3D) porous materials including carbon-based aerogels, foams and sponges with low density, good structural integrity and hydrophobicity have drawn increased attention interest as absorbents for wastewater treatments [4,5,6,7,8,9,10]. Among the carbon nanomaterials graphene oxide (GO) nanosheets, the hydrophilic counterpart of graphene, have shown great potential for integration into 3D monoliths due to the high surface area, ultralight weight and mechanical strength by varying solution methods [11,12,13,14,15].

The self-assembly of GO sheets is considered the most cost-efficient approach for the integration of GO into 3D monoliths by first forming a GO-based hydrogel via gelation of GO flakes and further subjecting it to a freeze-drying process to transform it into an aerogel by solvent sublimation. Thanks to decorating hydrophilic functional groups at both edges and basal planes, the GO tendency for aggregation and restacking could be exploited for such a purpose. 

In order to induce the self-assembly of the GO sheets, the functional groups and adsorbed water are removed and varying non-covalent forces arise, thus inducing the gelation process. Factors such as flake size, concentration, and solvent pH were reported for the control of gelation [16].

The solvothermal synthesis technique has been indicated as an efficient approach for the self-assembly of GO [17,18]. While the effects of GO and solvothermal conditions have been studied for the fabrication of amine-modified aerogels for CO_2_ capture, there are no results reported on the effect of GO properties as induced by oxidation conditions on the cost-efficient fabrication of good performance environmental absorbents for cleaning oil spills and treating organic solvents. The high temperature and pressure employed in a solvothermal process result in the removal of decorating oxygen functional groups and dehydration; thus, gelation takes place by aggregation and restacking interactions [12,19]. However, such high temperatures and pressure are not appropriate for scalable synthesis.

On the other hand, GO gelation by the chemical reduction has the advantage of not requiring high temperature while it allows the control of the hydrogel shape by simply changing the reaction vessel [20]. Nevertheless, the self-assembly of GO into 3D monoliths at temperatures below 100 °C has seldom been reported [21,22].

Given that the traditional hydrogels and aerogels exhibit some drawbacks including poor mechanical properties and limited functional properties [23], increased efforts have been made for the self-assembly of GO sheets and to improve traditional hydrogels. Recent works [24] have shown that the cost-efficiency of solvothermal process can be improved in terms of reducing the operating temperature and duration by applying simultaneous chemical reduction with different reducing agents including ammonia, hydrazine, or vitamin C for speeding the gelation of the reduced GO sheets [25,26]. 

Beside enhancing the gel rate, the consolidation of the hydrogel is of paramount importance for the assembly of GO sheets into 3D monoliths. Moreover, the absorption properties of porous monoliths are known to be controlled by the internal structure and porosity which are dictated instead by the fabrication approach. Till present, the integration of GO sheets into 3D porous monoliths with controlled porosity, morphology, and reduction degree/hydrophobicity is still a challenging task while the relationship between hydrothermal conditions and the properties of corresponding GO aerogel still lacks understanding. Therefore, the development of a facile and feasible strategy to self-assemble the GO nanosheets is highly required. Under this aspect, it was observed that the effect of functional group content on the GO gelation has not been reported until now. 

This work proposes a combination of solvothermal/chemical-induced gelation for tailoring the properties of graphene aerogel absorbent by a simple, low-cost, and scalable solution process aiming to obtain low density and well consolidated monoliths. Varying parameters were employed to allow the evaluation of the effects of reduction and consolidation with the purpose of obtaining performant 3D aerogels for oil/organic solvent separation. To the best of our knowledge, this is the first report on the effect of surface chemistry of GO on its gelation by a combined chemical–thermal reduction approach and further on the properties and absorption capacity of the aerogels. The benefits of such an approach consist in the fact that the gelation can be easily controlled by dehydration and removal of functional groups in GOs by one single step procedure, thus beside performance, the cost and time-efficiency are pursued. The total oxygen content and functional group distribution in GO were adjusted by either oxidation procedures or by improving the oxidant penetration in the inter-layer space by using an expanded starting graphite material [27]. An environmentally friendly reducing agent (vitamin C, denoted with VC) was employed for chemical reduction simultaneously with solvothermal reduction. Temperature ranges below and above 100 °C were employed in order to control the solvothermal reduction while the chemical reduction degree was adjusted by the GO:VC w/w ratio. The results obtained in this work seek to improve the knowledge on the combined chemical/solvothermal reduction effects and the controllable performances of GO aerogels by investigating the structural differences of the prefabricated aerogels and their various uptake behaviours.

## 2. Materials and Methods

All the chemicals were reagent grade (Alfa Aesar, Sigma Aldrich, Valencia, Spain) and used as received. Aqueous slurries of GO nanosheets obtained from both graphite (GO) and expanded graphite (GOx) were provided by Graphenea (Donostia, Spain).

GO aqueous dispersions (2 mg mL^−1^) were obtained by bath ultrasonic treatment for 1 h. Further, hydrogels were obtained by solvothermal approach in the presence of VC as a reducing agent by employing varying GO:VC weight ratios and temperatures ranging from 65 °C to 165 °C. Dehydration was applied by freezing the hydrogel at −20 °C. The three-dimensional porous aerogels were obtained by freeze-drying the resulted hydrogels at −80 °C under a high vacuum at 0.05 mbar LyoQuest freeze-drier (Telstar, Madrid, Spain) with three-directional cooling with a rate of about 12 degrees min^−1^ followed by sublimation at 20 °C for 48 h at 0.015 mbar.

The aerogel volume was measured with a calliper with accuracy 0.05 mm (measurement error is estimated as ±10%.). The apparent density of the aerogels was calculated by dividing the weight to their volume. The X-Ray Powder Diffraction (XRD) measurements were performed on a 2D Phaser equipment (Bruker, Madrid, Spain) with Cu-Kα radiation working at 30 kV and 10 mA in order to check the crystalline structure of the graphite materials. The surface morphologies were obtained by scanning electron microscopy (SEM, JSM 6300 JEOL, Tokyo, Japan) equipped with Energy Dispersive X-Ray Spectrometer (EDS, Oxford Instruments, Bristol, UK) for elemental composition measurements. The presence of functional groups was assessed by means of Fourier-transform infrared spectroscopy (FTIR). The FTIR spectra were acquired on a FT/IR-6200 (Jasco) spectrometer in the spectral window of 4000–400 cm^−1^ in ATR mode. The structure and composition of GOs were studied by X-ray Photoelectron Spectroscopy (XPS) with a photoelectron spectrometer VG-Microtech Multilab 3000 (Thermo Fisher Scientific Inc., Waltham, MA, USA) and Raman spectroscopy on a Xplora spectroscope using a 532 nm laser (Horiba, Villeneuve d´Ascq, France). Contact angle measurements were performed on OCA 15 Plus instrument (Data Physics Instruments, Neurtek, Spain) at room temperature to quantify the aerogel wettability. The contact angle values were the average of at least five measurements using 6 µL water droplets at different positions on each sample, followed by calculation with the SCA20 software. 

In order to assess the pollutant absorption capacity (*Q*) of obtained graphene aerogels, engine oil with density of 0.852 g cm^−3^ and dichloromethane were selected as representative oil and as organic solvent pollutants. The aerogels were immersed into the adsorbates and allowed to reach saturation at room temperature. The absorption capacity (*Q*) at saturation was determined by measuring the mass of the samples before (*m*_0_) and after absorption (*m*_1_), by using the equation *Q = (m*_1_ − *m*_0_)/*m*_0_. All absorption experiments were repeated five times.

## 3. Results

Varying oxygen content and distribution in GO were induced by employing an expanded graphite material. The resulting GOx nanomaterial was characterized and referenced to the GO obtained from non-expanded graphite. Figure 1 depicts the effects of starting material on the structural, compositional and vibrational properties of the GOs. 

As evidenced from XRD spectra in Figure 1A, the raw graphite material exhibits the typical (002) diffraction peak around 26 degrees. By expanding the graphite, the (002) peak shifts towards a lower angle and presents an increased full width half maximum (FWHM) value. Upon oxidation, the raw graphite spectrum shows a shift of the typical (002) peak to a value around 10.9 degrees while the GOx obtained from the expanded graphite exhibits a higher angle shift and FWHM. 

The presence of oxygen functional groups in the GOs was evaluated by FTIR spectroscopy, as indicated by the spectra depicted in Figure 1B. The GO nanomaterial exhibited the phenolic C–O and epoxy C–O–C stretching vibrations at about 1030 cm^−1^ and 1217 cm^−1^; the O–H group was located at 1388 cm^−1^; the peaks at 1721 cm^−1^ and 3584 cm^−1^ were attributed to carboxylic C=O and O–H stretching vibrations, while the band around 3020 cm^−1^ was attributed to C–H bond stretching vibrations. The aromatic C=C bending of graphene was located at 1614 cm^−1^ [28]. One could observe that the GOx counterpart exhibits a more intense carboxylic peak at 1721 cm^−1^ and O–H at 1388 cm^−1^.

The effect of expanding the starting graphite material on oxygen content and distribution in GOs was evaluated by XPS analysis. The C/O ratio decreased from GO to GOx from 2.15 to 1.84 confirming that the expanding approach results in increased oxygen content [18]. As it can be observed in the spectra depicted in Figure 1C, the C1s peak was deconvoluted into three components for both GOs. The contributions to C1s were identified as carbon atoms in C=C/C–C bonds (peak located at cca. 284.6 eV), carbon atoms in C–OH/C–O–C bonds (peak located at cca. 286.5 eV) and carbon atoms in COOH bonds (peak located at cca. 288.5 eV). The XPS analysis indicated the C=C/C–C : C-OH/C–O–C : COOH content (%) changed from 43.2 : 49.3 : 7.5 to 31.6 : 60.9 : 7.6.

Raman spectroscopy was employed to further evaluate the disorder degree in the GO nanomaterials. The G band characteristic to sp^2^ graphene lattice is identified by the peak at 1588 cm^−1^ and 1591 cm^−1^ for the GO and GOx, respectively, while the D-band generated by defects such as functional groups, holes, folding, etc. of the graphene layers, which is identified by the peak at 1349 cm^−1^ and 1352 cm^−1^, respectively. The Raman spectra of GO and GOx also show intensity ratios of the D and G bands (ID/IG) of 0.87 and 0.8, respectively.

In order to analyze the effect of the chemical reduction and hydrothermal reduction on the formation of hydrogels and aerogels, varying parameters were investigated. Thus, for a predominant chemical reduction, a higher amount of VC was employed while the temperature was kept below 100 °C, and for a predominant thermal reduction involving dehydration and removal of oxygen groups, a lower amount of VC was employed above 100 °C. 

The effect of solvothermal conditions on the volume of GO hydrogels obtained in the absence and presence of varying VC amount is depicted in Figure 2A for both GO nanomaterials. It can be observed that the hydrogel volume decreases with temperature and duration in both GO cases. However, increasing the temperature above 140 °C appears to induce only little change in the volume of the hydrogels. The hydrogel volume further decreases upon the addition of a reducing agent. On the other hand, the hydrogel volume appears higher for GOx counterparts as the GOx flakes are decorated with a higher number of oxygen groups and are less stacked, which results in better dispersed flakes. The VC appears to exhibit a stronger effect on the GOx counterparts below 120 °C.

The hydrogels were subjected to freeze-drying, and the density evolution of the obtained aerogels is shown in Figure 2B. Increasing temperature results in increased density values, which are in agreement with the decreased volume for the hydrogels depicted in Figure 2A. The density of the aerogels obtained above 120 °C is higher for the GOx aerogels and it is enhanced with the VC amount. Figure 2C depicts demonstrative images of the aerogels/hydrogels in different synthesis conditions and as a function of GO type. As it can be observed, the differences in volume for the GO and GOx aerogels, respectively, are in high agreement with the volume evolution in Figure 2A.

The morphological analysis of the obtained aerogels revealed a strong dependence on the synthesis conditions as observed in the exemplificative SEM images depicted in Figure 3. On one hand, by comparing the images in Figure 3A,B depicting the morphology of the aerogels obtained by predominant chemical reduction (low temperature range), the use of expanded graphite appears to induce the formation of 3D monoliths with thinner, less stacked and parallel partially reduced GOx sheets (see the areas evidenced by circles). The same observation is made by comparing the images in Figure 3C,D, with respect to the effect of GO synthesis conditions. On the other hand, the predominant chemical reduction (Figure 3A,B) appears to induce a structure with larger pores and larger constituent sheets than the consolidation induced by predominant thermal reduction (Figure 3C,D), but less homogeneous.

The reduction of the GO nanosheets composing the aerogels was assessed by EDS analysis and FTIR spectroscopy as depicted in Figure 4A,B, respectively. Due to the higher content of oxygen, the GOx counterparts exhibit a lower C/O ratio irrespective of the synthesis conditions. The VC effect is more obvious at low temperature consolidation of hydrogel/aerogel (below 100 °C), where the GOx aerogel exhibits a C/O ratio inferior to GO at the same amount of VC, while by increasing the temperature and duration to 120 °C 12 h, the C/O ratios of the obtained aerogels increase. It can be observed that by changing the synthesis conditions from predominant chemical reduction (GO:VC 1:8, 65 °C, 2 h) to predominant solvothermal reduction (GO:VC 1:2.5, 120 °C, 12 h), the C/O ratio of GO-based aerogels increases from 1.364 to 2.87 while for GOx ones it increased from 1.077 to 2.77.

Representative FTIR spectra recorded for the aerogels obtained in varying conditions (Figure 4B) present similar features but with lower intensity and slightly shifted towards increased wavenumber with respect to parent GO nanomaterials as previously depicted in Figure 1B. Amongst the functional groups, the phenolic C–O and carboxylic C=O located at around 1030 and 1721 cm^−1^ appear most affected by the coupled chemical–thermal reduction approach, in both cases of GO types. Moreover, by comparing spectra a and b of the aerogels obtained at a temperature below 100 °C and with 1:8 w/w ratio of reducing agent, one can observe less intense peaks in spectrum a (corresponding to GO aerogel). By comparing the spectra b and c which correspond to GOx aerogels obtained with GO:VC 1:8 and 1:2.5, respectively, one can observe less intense peaks in spectrum b. When the temperature is raised above 100 °C, e.g., 120 °C, one can observe clearly decreased peaks at 1726, 1418 and 1037 cm^−1^ and the band around 3400 cm^−1^ in spectra d. Moreover, it is evident that the peak located at 1726 cm^−1^ which is attributed to carboxylic groups is even less intense in spectrum d than in spectrum b.

The effect of synthesis conditions on the Raman bands is further depicted in Figure 4C. While all aerogels exhibit higher ID/IG values with respect to the corresponding parent GO nanomaterials, the disorder degree decreases by decreasing VC amount due to less molecules of reducing agent being available for reaction with GO flakes. The band intensity ratio lowers even more by increasing the temperature to 120 °C. 

The contact angle measurements were performed in order to assess the wettability properties of the obtained aerogels and representative images are further depicted in Figure 4D. On one hand, it was observed that the GOx aerogels exhibited higher contact angles than the GO counterparts. On the other hand, the consolidation by predominant chemical reduction results in lower contact angles than the aerogels obtained by predominant solvothermal reduction. 

The applicability of obtained aerogels for oil/water separation was tested as a function of synthesis conditions and type of GO nanomaterial. As it is observed in Figure 5A, the oil uptake increases with temperature for a given GO:VC w/w ratio. Such an increase is more evident for GO aerogels than the GOx ones. The aerogels obtained by predominant solvothermal reduction appear to exhibit better oil uptake than predominant chemical reduction, the oil capacity reaching 56.06 and 85.86 g g^−1^ for GO and GOx aerogel, respectively. The organic solvent separation results depicted in Figure 5B follow the same trend but reach higher values in all cases. The GOx counterparts exhibit higher performance than the GO ones. The organic solvent uptake capacity is higher for the aerogels consolidated by predominant thermal reduction, reaching more than two-fold the predominant chemical one. The demonstrative recording of the oil uptake by the aerogel obtained from GOx nanomaterial in conditions of predominant solvothermal reduction can be found in the supplementary video material. It can be observed that the aerogel presents itself as a light 3D monolith with a high absorption rate.

## 4. Discussion

It is known that oxidation conditions govern the GO properties. One important parameter in inducing the surface chemistry of GO is the starting graphite material. The effect of expanded graphite on the properties and oil/organic solvent uptake was analyzed. 

The characteristics of GO nanomaterials are depicted in Figure 1. The XRD results in Figure 1A indicated improved oxidation degree and lower stacked sheets number for GOx than the GO nanomaterial (about five with respect to 10 stacked sheets). On the other hand, the improved oxidation degree in GOx counterparts is confirmed by the more intense FTIR peaks and slight shifts of the carbonyl C=O stretching vibration, O–H and carboxylic C=O vibration at 1721, 1217 and 1388 cm^−1^, respectively. The XPS results indicated an increased oxygen content upon using an expanded graphite which is in agreement with XRD and FTIR results. A difference in oxygen group contributions is observed, where the use of expanded graphite results in higher C–OH/C–O–C contribution. Such results could be explained by an enhanced penetration of the oxidant into the inter-layer space between the expanded graphite sheets, thus leading to a higher contribution from C–OH/C–O–C on the basal graphene plane. On the other hand, the Raman spectroscopy confirmed varying defects introduced in the graphene framework by expanding the graphite. It has been suggested that despite the use of the ID/IG ratio for evaluating the disorder degree, no distinction can be made between holes and sp^3^ defects [29,30]

Individual GO sheets are subject to electrostatic repulsion from the functional groups and to van der Waals interaction between the basal planes when reduced in a solution [31]. The following processes take place simultaneously in the solvothermal synthesis of graphene aerogels: The removal of oxygen groups in GO (reduction) due to thermal effects and chemical reduction with VC and dehydration by removal of adsorbed water. In the following section, the results on the effects of a combined approach are presented. The analysis of temperature, duration and GO:VC w/w ratio on the hydrogel volume depicted in Figure 2 indicate the removal of an increased number of oxygen groups with temperature and duration in both GO cases. However, the constant volume of the hydrogels obtained above 140 °C indicates a limitation in the removal of oxygen groups. The volume decrease upon the addition of reducing agent is attributed to a combined effect of dehydration and chemical and thermal reduction. The VC appears to exhibit a stronger effect on the GOx counterparts below 120 °C which can be explained by an improved access of VC molecules towards the oxygen groups decorating the GOx sheets. The increase in density with temperature is attributed to improved thermal reduction and dehydration of the corresponding GO flakes. The addition of VC results in a combined effect of dehydration and thermal–chemical reduction. Thus, the higher density of GOx aerogels is attributed to better dispersed GOx flakes.

The less stacked GOx sheets in Figure 3 are induced by the improved oxidation of expanded graphite, which is in agreement with XRD results. The aerogels consolidated by predominant thermal reduction exhibit a more homogeneous structure with smaller pores due to the faster kinetics of thermal reduction with respect to chemical reduction.

The inferior C/O ratio of the GOx aerogel with respect to the GO one for the same amount of VC under low temperature (Figure 4A) can be explained by the inability of VC to remove the higher content of oxygen groups in GOx. On the other hand, the increase in temperature and duration result in increased C/O ratios of the obtained aerogels due to combined reduction and dehydration. 

All FTIR spectra in Figure 4B show reduced peaks and slightly shifted with respect to parent GO nanomaterials, confirming the partial removal of oxygen groups. Both types of GO-based aerogels show less intense peaks attributed to phenolic C–O and carboxylic C=O in comparison to parent GOs, which indicate that the combined reduction approach affects primarily these functional groups. The less intense peaks in the spectrum of GO aerogel vs GOx are attributed to a lower initial content of oxygen groups in GO nanomaterial. As the GOs sheets are exposed to a higher amount of VC, less intense peaks appear in FTIR spectra. The increase in temperature above 100 °C results in less intense peaks at 1726, 1418 and 1037 cm^−1^ and the band around 3400 cm^−1^ in the spectra d due to the removal of adsorbed water and oxygen groups by solvothermal reduction. The solvothermal reduction appears to result in marked removal of carboxylic groups with respect to the chemical one. 

The partial reduction of GO sheets in the aerogels is confirmed by the Raman results in Figure 4C. The intensity ratio obtained for the aerogels synthesized at temperature up to 120 °C indicate that the graphene structure is not affected by the removal of adsorbed water molecules. On the other hand, the GOx aerogel counterparts exhibit lower band intensity values which are attributed to the nature of graphite employed for the synthesis of GOx nanomaterial and less defective material.

Upon the reduction in solution, various regions of GO nanosheets become hydrophobic due to the restoration of sp^2^ domains and the removal of oxygen groups. The contact angle measurements in Figure 4D indicate that the GOx aerogels having ordered thinner sheets of reduced material exhibit higher contact angles than the GO counterparts. The aerogels obtained by predominant chemical reduction exhibit lower contact angles due to less oxygen groups that are removed, which is in agreement with FTIR results. Moreover, while VC addition is low, the increase in consolidation temperature results in an increased contact angle due to improved dehydration and removal of oxygen groups by thermal reduction.

The oil uptake results in Figure 5A show that the reduction degree induced by increasing temperature for a given VC affects the absorption capacity for both GO types of aerogels. On the other hand, the predominant solvothermal reduction yields the highest uptake reaching 101 cm^3^ g^−1^ of absorbed oil which may be accounted for by the homogeneity of the aerogel structure and contact angle values. The absorption rates indicated in Figure 5A show the best oil absorption rates reached 3.8 g s^−1^ (equivalent to 4.5 cm^3^ s^−1^) for the corresponding aerogel. 

The organic solvent uptake capacity depicted in Figure 5B follows the same trend as the oil one. The GOx aerogels exhibit improved performance for organic solvent separation as well, and such results are attributed to the enhanced dispersibility of original GO sheets and the homogeneity and ordered structure of the obtained aerogel. The best performance in terms of organic solvent absorption rate reached 6.3 g s^−1^ 8 (equivalent to 4.7 cm^3^ s^−1^ of absorbed solvent) for the aerogels obtained by consolidation through the predominant solvothermal reduction.

## 5. Conclusions

The fabrication of low-density GO aerogels by a combined solvothermal–chemical reduction approach is reported. The effects of various parameters such as reducing agent, temperature, duration, and GO oxidation degree on the properties of corresponding aerogels are evaluated. It is shown that the use of expanded graphite induces higher oxygen content and a larger contribution from C–OH/C–O–C in the GOx nanomaterial that translates into higher hydrogel volume and further in an aerogel with more homogeneously distributed sheets that induce improved porosity. The oil and organic solvent uptake measurements indicated the GOx aerogels exhibited the best performance. The gelation by adjusting the combined chemical and solvothermal reduction with the predominant thermal removal of oxygen groups results in the best uptake values. Therefore, the enclosed results demonstrate the viability of employing the chemistry of GO for adjusting the gelation of GO sheets into hydrogel by combined chemical/solvothermal approach towards improving the oil and organic solvent absorption properties of graphene aerogels.

## Figures and Tables

**Figure 1 nanomaterials-09-01077-f001:**
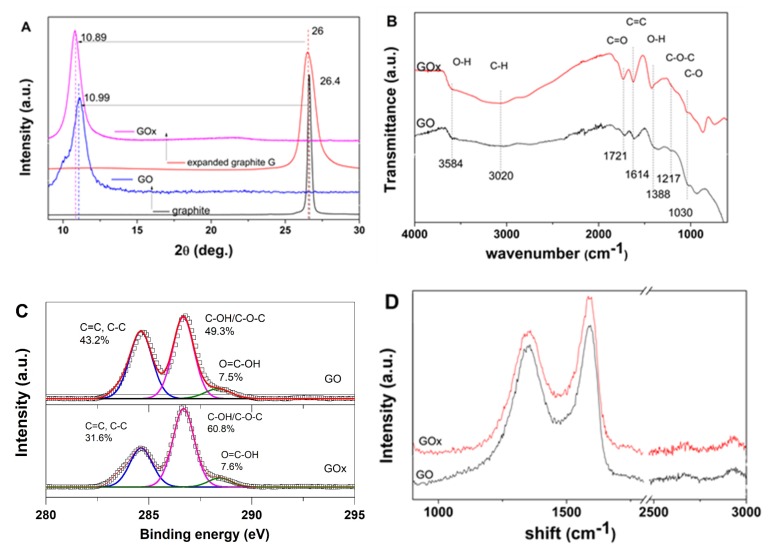
XRD spectra of graphite materials and resulting GOs (**A**), FTIR spectra (**B**), C1s high resolution spectra (**C**) and Raman spectra (**D**) of GO nanomaterials.

**Figure 2 nanomaterials-09-01077-f002:**
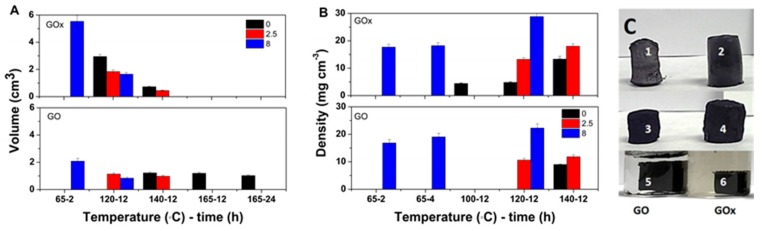
Hydrogel volume (**A**) and aerogel density (**B**) as a function of GO:VC w/w ratio and temperature; Digital images of aerogels from GO and GOx (**C**) obtained with 1:8 GO:VC @ 65 °C for 4 h (1 and 2), obtained with 1:2.5 GO:VC @ 120 °C for 12 h (3 and 4) and hydrogels with 1:2.5 GO:VC @ 140 °C for 12 h (5 and 6).

**Figure 3 nanomaterials-09-01077-f003:**
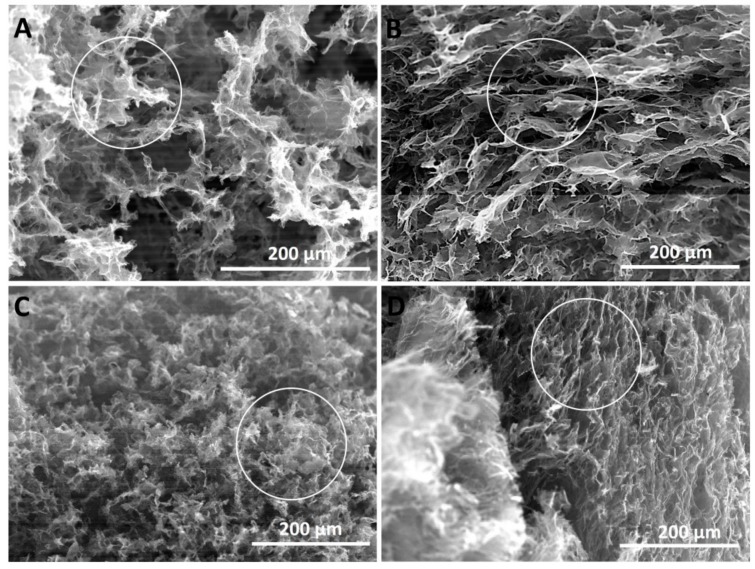
SEM images of GO and GOx with 1:8 GO:VC @ 65C@2 h (**A**,**B**) and 1:2.5 GO:VC @ 120C@12 h (**C**,**D**).

**Figure 4 nanomaterials-09-01077-f004:**
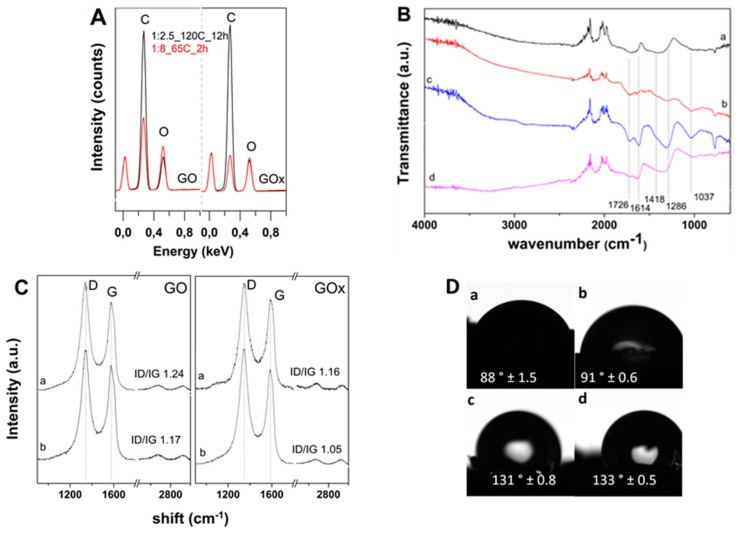
(**A**) EDS spectra of aerogels from GO and GOx dispersions; (**B**) FTIR spectra of aerogels from GO and GOx at 1:8_65 °C_4 h (a and b), GOx at 1:2.5_4 h at 65 °C and 120 °C (c and d); (**C**) Raman spectra of aerogels from GO and GOx at 1:8_65 °C_2 h (a) and 1:2.5_120 °C_12 h (b); (**D**) Average contact angle of GO (left) and GOx (right) aerogels obtained in the conditions: 1:8_65 °C_4 h (a and b), and 1:2.5_120 °C_12 h.

**Figure 5 nanomaterials-09-01077-f005:**
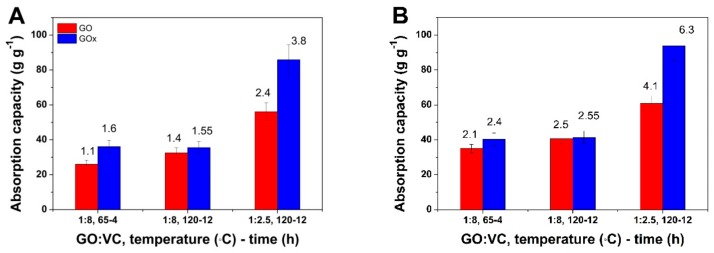
Adsorption capacity evolution and the absorption rate (g s^−1^, values indicated in the plot) for GO and GOx aerogels for oil/water (**A**) and organic solvent (**B**) separation.

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
