# Peer review of "Tailoring the Performance of Graphene Aerogels for Oil/Organic Solvent Separation by 1-Step Solvothermal Approach"

_nanomaterials, 2019, doi:10.3390/nano9081077_

Round 1
Reviewer 1 Report
Please allow me to write a review for this very attractive, interesting and valuable manuscript which is very suitable to “nanomaterials” reporting great investigation ranges associated with “the fabrication of low density GO aerogels by a combined solvothermal-chemical reduction approach”. The contents in this paper are certainly novel and the quality of the work is high. The technical validity of the characterization work is good, as to XRD, FTIR, XPS, Raman, SEM and contact angle measurements. I feel that this very valuable article deeply demonstrates the viability of employing the interesting chemistry area of GO for adjusting the gelation of GO sheets into hydrogel by well-setting chemical/solvothermal approach methods. I can confirm that the characterizations appear to be very well carried out. I am very happy to recommend publication as original paper in the “nanomaterials”. The authors may wish to take into account the following points when putting together a final, publishable, version of the manuscript.
1) As to Figures 1 and 4, would the authors please describe in details how spectral assignments are for valuable peaks?
2) Differences between this paper and ref.17, ref 18 should be much clearer.
3) Would the authors please describe in details what and how important “1-step solvothermal approach” is in this paper?
4) How is Raman spectroscopy very powerful tool for the nanomaterials in this paper?
5) As to Figure 5, do values have any errors?
Reviewer 2 Report
In this manuscript by Pruna et al. the authors report preparation of graphene aerogels prepared using a combination of solvothermal and chemical induced gelation resulting in aerogels of variable characteristics. The aerogels are then characterized with a suite of techniques and investigated for their ability to absorb engine oil and organic solvents. The absorption properties are found to be better for the aerogels with low C/O ratio. These aerogels may be used for cleaning oil spill and treating organic solvents. As such the aerogels prepared and the relevant study is of practical interest and qualifies for publication in the journal Nanomaterials. However there are some issues (listed below) that the authors should address before this work is accepted for publication.
The authors have quantified the aerogels prepared for their wettability by measuring contact angle. However, the authors have not specified what wetting fluid have they used for this purpose. Without this specification, contact angle measurements do not carry much meaning.
On page 5, describing Figure 3, the authors state, 'On one hand the use of expanded graphite appears to induce the formation of 3D monoliths with thinner, less stacked and parallel partially reduced GOx sheets...' While this conclusion can indeed be made from other data, in my opinion the SEM images do not seem to show this clearly. If indeed they show this, perhaps the authors can mark the relevant regions in the images from which this conclusion can be drawn. This would help the reader.
Describing Figure 4B, on page 6, the authors state, 'By comparing the spectra a and b of the aerogels obtained...one can observe less intense peaks in spectrum a...' To me, on the contrary, spectrum b appears to have less intense peaks.
In the discussion section, all the figures are described again resulting in repetition of some information that is already given in the results section. It would be better to avoid this and limit this section only to new information obtained by making inferences from the pervious section.
In the discussion section, the authors provide values of absorption rates for both oil as well as organic solvent absorption by the aerogels. I think this is an important piece of information and can help quantify another aspect of absorption capability of the aerogels. For this reason, I suggest including this data for all the samples investigated in the results section. Currently, the authors state only the maximum rates.
Finally, the authors have provided a video of (perhaps) an aerogel absorbing a dyed liquid. However, no information is provided about this video in the manuscript. This supplementary material is interesting and should be described in more details in the manuscript.
